# Genetic Markers for Thrombophilia and Cardiovascular Disease Associated with Multiple Sclerosis

**DOI:** 10.3390/biomedicines10102665

**Published:** 2022-10-21

**Authors:** Maria S. Hadjiagapiou, George Krashias, Elie Deeba, George Kallis, Andri Papaloizou, Paul Costeas, Christina Christodoulou, Marios Pantzaris, Anastasia Lambrianides

**Affiliations:** 1Department of Neuroimmunology, The Cyprus Institute of Neurology and Genetics, P.O. Box 23462, 1683 Nicosia, Cyprus; 2Department of Molecular Virology, The Cyprus Institute of Neurology and Genetics, P.O. Box 23462, 1683 Nicosia, Cyprus; 3Karaiskakio Foundation, P.O. Box 22680, 1523 Nicosia, Cyprus

**Keywords:** multiple sclerosis, inflammation, thrombophilia, cardiovascular disease (CVD), expanded disability status scale (EDSS), multiple sclerosis severity score (MSSS)

## Abstract

Multiple sclerosis (MS) is an autoimmune inflammatory disease of the central nervous system (CNS) with an unknown etiology, although genetic, epigenetic, and environmental factors are thought to play a role. Recently, coagulation components have been shown to provide immunomodulatory and pro-inflammatory effects in the CNS, leading to neuroinflammation and neurodegeneration. The current study aimed to determine whether patients with MS exhibited an overrepresentation of polymorphisms implicated in the coagulation and whether such polymorphisms are associated with advanced disability and disease progression. The cardiovascular disease (CVD) strip assay was applied to 48 MS patients and 25 controls to analyze 11 genetic polymorphisms associated with thrombosis and CVD. According to our results, FXIIIVal34Leu heterozygosity was less frequent (OR: 0.35 (95% CI: 0.12–0.99); *p* = 0.04), whereas PAI-1 5G/5G homozygosity was more frequent in MS (OR: 6.33 (95% CI: 1.32–30.24); *p* = 0.016). In addition, carriers of the HPA-1a/1b were likely to have advanced disability (OR: 1.47 (95% CI: 1.03–2.18); *p* = 0.03) and disease worsening (OR: 1.42 (95% CI: 1.05–2.01); *p* = 0.02). The results of a sex-based analysis revealed that male HPA-1a/1b carriers were associated with advanced disability (OR: 3.04 (95% CI: 1.22–19.54); *p* = 0.01), whereas female carriers had an increased likelihood of disease worsening (OR: 1.56 (95% CI: 1.04–2.61); *p* = 0.03). Our findings suggest that MS may be linked to thrombophilia-related polymorphisms, which warrants further investigation.

## 1. Introduction

Multiple sclerosis (MS) is a chronic, autoimmune inflammatory disease of the central nervous system (CNS), characterized by demyelination and neurodegeneration; however, a growing body of evidence indicates the association of chronic, autoimmune disorders with the onset of cardiovascular disease (CVD), development of deep vein thrombosis (DVT), and pulmonary embolism (PE) [1,2]. In fact, a high risk of ischemic stroke, coronary artery disease, and venous thromboembolism (VTE) has been reported in patients with MS compared with the general population, especially during the early onset of the disease [3,4,5].

Recently, it has been discovered that VTE and CVD are caused by the interplay of environmental and genetic influences, identifying genetic polymorphisms that predispose for cardiovascular perturbation and prolonged thrombin formation [6].

One of the factors associated with VTE is a point mutation in the gene encoding the factor V (FV) that substitutes guanine for adenine (G>A) at nucleotide 1691, resulting in a missense mutation, also known as FV Leiden R506Q [7]. FV is the cofactor of activated FXa for the generation of thrombin and is negatively regulated by activated protein C (APC). The substituted 506 amino acid region constitutes one of the sites on the FVa molecule that the APC recognizes and binds to cleave the FVa protease. Thus, the presence of the mutation FV Leiden (R506Q) renders FVa resistant by up to 90%, suppressing the anticoagulant function of APC and promoting FXa–FVa complex formation and increased thrombin generation [7]. Interestingly, compound heterozygosity of FV Leiden and prothrombin mutation (G20210A) has also been reported as a critical thrombotic risk mediator with more than 40-fold increased risk for VTE [7]. The prothrombin G20210A variant has been found to increase both mRNA accumulation and protein synthesis, whereas its presence enhances approximately threefold the relative risk for thrombosis [8].

Another thrombosis-related mutation has been identified at nucleotide 4070 in the FV gene (FV R2, FV H1299R), and even though this variant does not have a significant impact on the clinical outcome, it can increase resistance to APC either in homozygous status or compound heterozygosity with FV Leiden [9].

The interaction between coagulation and inflammation has been reported in inflammatory and neurodegenerative diseases of the CNS, in which thrombin and fibrin(ogen) play a crucial role in triggering demyelination and neuroinflammation. Mutations in the fibrinogen gene, particularly the transition from guanine to adenine at nucleotide 455 of the beta-fibrinogen promoter, lead to elevated fibrinogen levels. Furthermore, the −455 G>A variant has been identified as a risk factor for fibrin clot formation in PE cases [10] and predisposes to coronary artery disease, as well as cerebral infarction [11,12]. So far, in MS animal models, thrombin and fibrin(ogen) have been detected in pre-demyelinated CNS areas, attracting resident and infiltrating immune cells from the periphery, contributing further in abnormal BBB permeability, and stimulating the inflammatory process [13]. Consequently, analysis of the beta-fibrinogen −455 G>A should be considered with respect to the fibrinogen accumulation, since −455 G>A mutation could be associated with the abnormal deposition of elevated fibrinogen into the CNS, initiating inflammatory responses.

Genetic alterations have also been detected in the gene encoding the plasminogen activator inhibitor-1 (PAI-1), an essential inhibitor of the tissue plasminogen activator (tPA), leading to the suppression of plasminogen conversion to plasmin, interfering with fibrin lysis [14]. PAI-1 4G/5G polymorphism constitutes the most significant variant identified in the promoter region at nucleotide position 675 of the PAI-1 gene, and it is a guanine insertion or deletion (4G/5G) correlated with alterations in the transcription rate of the PAI-1 molecule [15]. While both 4G and 5G alleles have a transcriptional activator binding site, 5G has a transcriptional repressor binding site, resulting in a decreased transcription rate [14,15]. On the other hand, individuals homozygous for the 4G allele have an increased transcription rate and thus higher plasma levels of PAI-1 than the general population [16]. PAI-1 has been implicated in the inhibition of extracellular proteolysis and the suppression of fibrinolysis, promoting further the generation of fibrin clots, leading to thrombosis [15].

Fibrin clots are stabilized by the FXIII, a known zymogen that is activated by thrombin-mediated cleavage at amino acid residue 37 and cross-links fibrin [17]. In exon 2 of FXIII, the FXIII Val34Leu variant appears to provide increased activation but does not increase the plasma concentration levels of FXIII [17,18]. Paradoxically, the expression of this particular variant confers significant protection against CVD and VTE while affecting fibrin clot stabilization. In fact, the fibrin that cross-links with FXIII Val34Leu has smaller pores, thinner fibers, and altered findings of permeation [18,19].

Patients with MS have also shown lower folic acid levels and higher homocysteine concentration in plasma and cerebrovascular fluid (CSF) compared with healthy individuals [20]. A growing body of evidence supports the fact that increased homocysteine levels cause neurotoxic effects as a result of oxidized sulfhydryl groups and the release of reactive oxygen species following the overstimulation of excitatory N-methyl-D-aspartate receptors (NMDARs) in the membranes of neuronal cells [21,22]. The NMDAR triggers the activation of the ERK/MAPK pathway and phosphorylates the ERK1/2 kinases in the presence of calcium ions. When ERK1/2 phosphorylation is inhibited, homocysteine-induced neuronal death is reduced, clearly demonstrating the interplay between homocysteine, NMDAR, and MAPK signaling pathways [21]. Neurotoxicity triggers neuronal susceptibility to DNA damage and cellular apoptosis and affects the structural stability of myelin, promoting the degradation of the myelin sheath [23]. Since homocysteine is metabolized to methionine by the donation of methyl groups, low levels of methionine result in abnormally elevated homocysteine levels, which, in turn, reduce methyl groups donation and suppress methionine remethylation [23]. The influence of the methionine on the myelin basic protein (MBP) is significant as it makes the myelin sheath more stable by MBP methylation and retains hydrophobic properties [23]. Therefore, deficiency or suppression of methionine is associated with unstable myelin structure and myelin sheath rupture [23]. Methylenetetrahydrofolate reductase (MTHFR), which regulates both folate and homocysteine levels, plays a crucial role in the methionine synthesis pathway and ultimately is essential for the donation of methyl groups during the remethylation of methionine [20,24]. Mutations in the MTHFR gene impair the activity of the enzyme and, therefore, disrupt the regulation of plasma folate and homocysteine concentration [25]. Noteworthy, individuals homozygous for the MTHFR 677 C>T showed decreased activity of the enzyme at a rate between 35% and 70%, whereas the heterozygosity of the compound MTHFR 677 C>T and MTHFR 1298 A>C leads to a decreased activity rate from 50% to 60% [24,26]. Overall, elevated homocysteine levels have been linked to neuronal toxicity, stroke, coronary artery disease, and thrombophilia in either mutation [25,27].

Furthermore, predisposition to stroke, myocardial infarction (MI), or coronary artery disease was correlated with human platelet antigen (HPA-1) 1a/b polymorphisms [28,29], whereas MI was further associated with mutations linked to the gene that codes the angiotensin-converting enzyme (ACE) [29].

Given the significance of variants in genes encoding molecules involved in the coagulation cascade and the importance of coagulation–inflammation interplay in neuroinflammatory and neurodegenerative diseases, the purpose of our research was intended to examine such variants in MS. Specifically, we aimed to examine whether patients with MS displayed overrepresentations in the following genetic markers: FV Leiden polymorphism, FV R2, Prothrombin 20210 G>A, FXIII Val34Leu, beta-fibrinogen −455 G>A, HPA1a/1b, MTHFR 677 C>T, MTHFR 1298 A>T, PAI-1 4G/5G, ACE I/D, and Apo B R3500Q compared with healthy individuals. Moreover, we investigated whether such polymorphisms can serve as predictors of advanced disability and progression of the disease.

## 2. Materials and Methods

### 2.1. Study Participants

Forty-eight patients with MS were recruited into the study from the Department of Neuroimmunology at the Cyprus Institute of Neurology and Genetics between June 2021 and September 2021. The participants met all McDonald’s revised criteria for inclusion [30] and strictly adhered to the following guidelines: age above 18 years old; patients with clearly identified classification (Relapsing–Remitting MS (RRMS), Secondary Progressive (SPMS), Primary Progressive MS (PPMS)); and sufficient demographic and clinical data (age of onset, expanded-disability status scale (EDSS), medication, other autoimmune and non-autoimmune disorders). The exclusion criteria were as follows: whether the patient is pregnant or has a history of drug and alcohol abuse.

The study also included 25 healthy controls (HCs) matched for sex with the patients selected. Table 1 demonstrates the demographic, clinical, and laboratory characteristics of the participants. All the participants were thoroughly informed about the objectives of the current study and signed written consent form, approved by the Cyprus National Bioethics Committee (ΕΕΒΚ/ΕΠ/2016/51).

### 2.2. Genetic Analysis for Thrombophilia and Cardiovascular Risk Factors in MS Patients

Fresh blood samples were collected in vacutainers containing ethylenediaminetetraacetic acid (EDTA) following the extraction of genetic material using a QIAamp DNA blood mini kit (Qiagen, Venlo, The Netherlands).

CVD Strip Assay (ViennaLab, Wien, Austria) was applied for the detection of genetic variants associated with thrombophilia and CVD, following the manufacturer’s specifications [6]. The analyzed markers are presented in Table 2.

All gene sequences were concurrently amplified by polymerase chain reaction (PCR) and labeled by biotin as follows. The initial denaturation step was conducted at 94 °C for 2 min, followed by 35 cycles of 94 °C for 15 s, 58 °C for 30 s, 72 °C for 30 s, and 72 °C for 3 min for the final extension step. PCR products were then hybridized to a test strip containing sequence-specific oligonucleotide probes for the wild-type and mutant sequences. The bound biotin-labeled sequences were detected by a streptavidin-alkaline phosphatase substrate.

### 2.3. Statistical Analysis

Fisher’s exact test was used to assess sex matching between study groups, and the Hardy–Weinberg equilibrium test was used to examine whether frequencies in the control group matched the expected frequencies over time. The chi-square test, and, when necessary, Fisher’s exact tests were used to determine genotype and allele frequency distributions, whereas logistic regression analysis was applied to evaluate possible associations between different variables. Significant results were determined when the *p*-value was less than 0.05.

## 3. Results

### 3.1. Demographic and Clinical Data of Study Participants

Forty-eight patients with MS who provided data on both clinical and nonclinical features were recruited in the current research. According to the data indicated in Table 1, the average duration of the disease was 15.52 ± 8.5 years, whereas 50% of the MS participants had mild disability status (EDSS: 0–3.0), 35.4% had moderate (EDSS: 3.5–5.5), and 14.6% showed severe disability (EDSS: 6.0–9.5). Additionally, 8.3% of individuals were experiencing an aggressive disease progression (MSSS: 7–10).

The results of a sex-specific analysis of the disability status and MS severity score revealed an increase in EDSS and MSSS in men compared with women. In particular, male patients showed an EDSS median of 4.00 (interquartile range, IQR: 2.5–6.0) and MSSS median of 4.20 (IQR: 2.82–5.41), whereas female patients had an EDSS median of 3.0 (IQR: 2.5–4.75) and MSSS of 3.14 (IQR: 2.35–5.48). The patients were selected on the basis that they tested positive for at least one IgG antibody against coagulation components in our previous research work [31]. Table 1 shows the IgG antibodies that were analyzed, whereas detailed laboratory findings for each patient are provided in Appendix A.

Twenty-five HCs were also recruited for the current study, matching their sex with the selected MS patients (*p* = 0.8).

### 3.2. Analysis of Genotype and Allele Distributions of Thrombophilia and Cardiovascular Polymorphisms in MS Patients and Healthy Controls

The distribution of genotype and allele frequencies of polymorphisms related to thrombophilia and CVD are presented in Table 3, Table 4, Table 5 and Table 6. Genotypic frequencies for all loci tested in the control group satisfied the Hardy–Weinberg equilibrium.

Individuals homozygous for the FV Leiden, FV R2, and prothrombin G20210A polymorphisms were identified neither in the disease group nor the controls, and the same observation was found when analyzing the Apo B R3500Q variant.

Although there was no significant difference between MS patients and HCs for most of the tested mutations, a statistically significant difference was revealed in the genotype frequencies between MS patients and HCs for the FXIIIVal34Leu and PAI-1 4G/5G polymorphisms, as indicated in Table 3 and Table 4, respectively. More specifically, the genotype frequency of heterozygous FXIII V34L was significantly lower in MS patients compared with that in HCs (*p* < 0.05), and simultaneously, it was less associated with the disease occurrence according to the odds ratio (OR: 0.35 (0.12–0.99); *p* < 0.05). Moreover, the frequencies of PAI-1 polymorphisms were 16% for 4G/4G, 46% for 4G/5G, and 38% for 5G/5G in the patient group, whereas the corresponding frequencies in the healthy participants were 32%, 56%, and 12%, respectively. Following a comparison between 4G/4G and 5G/5G homozygous variants, the presence of 5G/5G appears to predominate in the MS group compared with controls (*p* = 0.02), and in regard to the 4G/4G genotype, the presence of the 5G/5G genotype was more likely to be associated with the disease (OR: 6.33 (95% CI: 1.32–30.24)) (Table 4). This finding was supported further by the analysis of the 5G/5G genotype against the 4G/4G + 4G/5G compound genotypes in both study groups, demonstrating a significant difference (*p* = 0.016) in terms of genotypic frequency, revealing once again that the homozygous 5G/5G genotype was correlated with the disease onset as opposed to the 4G/4G or compound 4G/4G + 4G/5G polymorphisms (OR: 4.80 (1.26–18.31); *p* < 0.05). Similarly, assessing the allele frequency, we observed that the 5G allele distribution was significantly frequent and more likely to be associated with the disease group compared with HCs (OR: 2.39 (1.19–4.81); *p* = 0.013) (Table 6).

### 3.3. Mutant Allele Presence and Disability Progression in MS

The results of the analysis of MS disability and progression revealed significant findings in terms of genetic variants related to thrombophilia and CVD (Table 7 and Table 8). Namely, the presence of the FV Leiden GA genotype was 47% less likely to be associated with advanced disability (OR: 0.53 (0.25–0.90); *p* < 0.05), and this was extended to the FV R2 AG, which had a 54% decrease in the likelihood of increased EDSS (OR: 0.46 (95% CI: 0.19–0.84); *p <* 0.001) and a 40% decrease in MSSS (OR: 0.60 (95% CI: 0.30–0.97)) (Table 7). Furthermore, the results of the analysis of the EDSS and MSSS median indices in regard to the FV polymorphisms demonstrated a lower median index being associated with the mutant genotype compared with the wild type (Table 7).

Interestingly, another significant correlation was observed when assessing the HPA-1a/1b genotype regarding the EDSS (*p* < 0.05) and MSSS (*p* < 0.05). Patients with HPA-1a/1b had a 47% possibility of having an increased disability score (OR: 1.47 (95% CI: 1.03–2.18); *p* < 0.05), whereas the results of the analysis of disease progression, according to the MSSS score, also showed a correlation with the presence of such genotype (OR: 1.42 (95% CI: 1.05–2.01); *p* < 0.05). In parallel, the median index of EDSS was higher in the presence of HPA-1a/1b (median index: 4.5 (IQR: 2.87–6.12)) compared with the HPA-1a/1a genotype (median index: 3.25 (IQR: 2.50–5.50)), which also holds true for MSSS (Table 7).

The results of a detailed analysis of HPA-1a/1b genotype frequencies indicated that there was a significant difference between male and female patients with MS. As shown in Table 9, male HPA-1a/1b carriers were significantly more likely to have a high EDSS score (OR: 3.04 (1.22–19.54); *p* < 0.05), whereas female carriers showed no significant association. On the other hand, female MS patients carrying the HPA-1a/1b genotype had an increased likelihood to be correlated with an increased MSSS score (OR: 1.56 (1.04–2.61), *p* < 0.05); however, such a correlation was not observed in the group of male patients.

Another important observation was revealed when analyzing the MTHFR A1298C polymorphism. Although there was no association between heterozygous or homozygous genotypes for the MTHFR A1298C polymorphism and the clinical outcomes of the disease, the results of a further analysis in male patients carrying the mutant C allele demonstrated an increased likelihood of having worse MS progression (OR: 1.66 (1.02–3.43), *p* < 0.05), a finding that was not observed in female patients (Table 9).

## 4. Discussion

MS is a chronic complex disease in which underlying mechanisms influenced by genetic, epigenetic, and environmental factors play a crucial role in disease onset. Although the etiology of MS is still under investigation, changes in the genetic makeup can affect the signaling pathways involved in the onset of MS. Previous studies have highlighted the importance of the interaction between coagulation components and inflammation [5,13,31,32], and, therefore, mutations in genes encoding procoagulant serine proteases or regulators of the coagulation cascade could be essential risk factors for MS.

In this case-control study, we analyzed different mutations identified in related genes to thrombophilia and CVD, and we examined their frequency and impact in patients with MS. In detail, we investigated the FV Leiden polymorphism, FV R2, prothrombin 20210 G>A, FXIII Val34Leu, beta-fibrinogen −455 G>A, HPA1a/1b, MTHFR 677 C>T, MTHFR 1298 A>T, PAI-1 4G/5G, ACE I/D, and Apo B R3500Q, and, to the best of our knowledge, this is the first study assessing such polymorphisms that may increase the predisposition of thrombotic episodes or cardiovascular events in MS, known pathological conditions in early disease occurrence [33,34].

The studied mutations in the FV gene, namely FV Leiden (FV R506Q) and FV R2 (FV H1299R), affect the sensitivity of the FV molecule to APC and disrupt the APC-mediated anticoagulant mechanism. These mutations prevent the binding of the APC to the FVa cleavage site, increasing the resistance to the APC and disrupting the anticoagulant mechanism [35]. In addition, the FV R2 mutation (FV H1299R) promotes the expression of the FV in plasma, which results in increased thrombin generation [36]. FV Leiden is a major risk factor for inherited thrombophilia and is identified in 20–50% of individuals with VTE, whereas the FV R2 mutation increases the risk of VTE and CVD in the presence of FV Leiden. Researchers found that Greek Cypriots from the general population showed heterozygosity of FV Leiden and FV R2 at frequencies of 7% and 10%, respectively [6]. For both mutations, no homozygous cases were identified. In other studies, assessing different ethnic groups, the frequency of heterozygous FV Leiden genotype ranged between 2% and 4% in healthy individuals [29,37], whereas in patients with coronary artery disease, it was 3.7% [29]. Furthermore, a frequency of 9.5% to 16.5% for FV R2 heterozygosity was shown in patients with VTE, in contrast to 5.8% to 10.4% in the general population, having a fluctuation in the proportion due to different ethnicities and sample sizes [38,39,40]. Our findings were in accordance with previous studies, showing that no homozygous individuals were identified in either the disease group or the control group, revealing similar heterozygous genotype frequencies as well as allele distribution in both study groups. Based on a comprehensive analysis of our findings regarding disability status and disease worsening, it was found that MS patients carrying heterozygous FV Leiden and FV R2 mutations had a reduced risk of developing advanced disability and were less likely to progress to advanced stages of the disease. As far as we know, this is the first study that aimed to correlate the FV Leiden and FV R2 vascular risk factors with clinical outcomes in MS and to deal with their potentially harmful effects on disease progression.

Importantly, FV Leiden carriers are four times more likely to experience recurrent thrombosis, and this finding is further extended to those individuals carrying the compound FV Leiden and prothrombin G20210A mutations, as they are 40% more prone to recurrence of thrombosis and having the first episode under the age of 45 years [7,41]. The prothrombin G20210A variant is correlated with elevated prothrombin levels in plasma, leading to an increased rate of thrombin generation and activation [42]. Studies in patients with coronary artery disease found a low frequency (3.3%) of the prothrombin G20210A heterozygous genotype, whereas the homozygous genotype was not identified [29]. Similarly, patients who developed thromboembolic episodes showed low frequency for the G20210A heterozygous genotype, and no homozygous individual was identified [7]. The findings of our study were consistent with other studies as we detected only one heterozygous case for the prothrombin G20210A variant in the entire cohort of MS participants. In addition, we found no healthy individuals being heterozygous or homozygous for such mutation, further confirming studies from European and non-European countries, in which the G20210A polymorphism is rarely detected in the general population [6].

Interestingly, when analyzing the genotype distribution of FXIII Val34Leu mutation between patients and healthy controls, we found a significant linkage between wild-type FXIII and the disease. Conversely, the heterozygous model of such polymorphism was associated with the control individuals rather than the disease group. It has previously been suggested that FXIII Val34Leu provides a protective effect against thrombosis as it affects the cross-linked structure of fibrin clots and suppresses the mass of thrombi [19,43]. Other researchers reported a high allele frequency, ranging between 21% and 28% for the Val34Leu variant in patients with VTE. Furthermore, they speculated that the Val34Leu variant in the presence of mutations known to promote VTE development, such as FV Leiden and prothrombin G20210A, might disrupt the mechanisms underlying VTE development [6,44]. Our results are consistent with such observations from previous research, highlighting the importance of FXIII in the study of neuroinflammatory diseases like MS.

The results of the analysis of the HPA-1 polymorphism showed differences when assessing disability and disease progression. Based on logistic regression analysis, the HPA-1a/1b genotype was more likely to be associated with increased EDSS and MSSS scores, whereas the HPA-1a/1a genotype was less likely to be associated.

Recent studies highlighted the fact that male MS patients annually experienced a greater deterioration in disability status compared with female patients, and, at the same time, male patients also exhibited disease worsening, including atrophy of the gray matter [45,46]. With regard to the HPA-1 polymorphism, our study showed significant findings due to the sex differences in MS. In comparison with male MS patients with the HPA-1a/1a genotype, male MS patients with the HPA-1a/1b genotype showed a significant association with advanced disability status; however, such an association was not demonstrated for female MS patients harboring the 1b allele. Furthermore, the results of our analysis indicated that in female patients carrying the HPA-1a/1b genotype, there was an increased likelihood of disease progression; however, this was not true for male patients carrying the 1b allele. In view of these findings, it is increasingly clear that sex plays a role in MS severity, even in genetic predispositions that show significant sex differences.

Substitution of proline for leucine at position 33 of the extracellular beta-3 domain of aIIb3 platelet integrin results in a change from HPA-1a to HPA-1b and disrupts the balance of local structure, thereby enhancing the adhesion of platelets to fibrinogen, increasing the flexibility of integrin, and the risk of thrombosis [47]. Our findings support the notion of the HPA-1b contribution to the cross-talk between thrombosis and inflammation, and further studies are required to investigate its role in neuroinflammatory diseases. In addition, the results of the analysis of the HPA-1 polymorphisms in the groups studied showed that only HPA-1a/1a and HPA-1a/1b genotypes were detected, whereas nonhomozygous individuals for the 1b allele were identified. Earlier studies in patients with coronary artery disease reported the presence of HPA-1a/1a and HPA-1a/1b genotypes in the study participants, whereas only HPA-1a/1a was identified in the control group [29]. In contrast, screening Greek Cypriots from the general population showed the presence of the HPA-1b/1b variant at a frequency of 4%, and this supports the notion of further analysis of such polymorphisms to better validate the results [6]. Interestingly, the 1b allele has been associated with the early onset of MI and ischemic stroke, and this has been demonstrated by a large case-control study including 1211 patients from Germany and 510 patients from the U.S.A. with coronary artery disease in which participants carrying the 1b allele had an increased risk of early onset of MI [29].

Moreover, our results showed no significant difference when assessing the frequency distribution of the beta-fibrinogen (−455 G>A) genotype in the studied groups or the association between beta-fibrinogen (−455 G>A) and disability or disease progression.

Consistent with our findings, a large investigation involving 426 patients who had suffered an ischemic stroke and 234 healthy volunteers showed no difference in the frequency of the beta-fibrinogen 455 G>A mutation between the study groups [29]. In addition, a study on MS found a weak association between the −455 G>A mutation and disease, further supporting a weak contribution to disease susceptibility [48]. Nonetheless, homozygosity for −455 G>A represents an increased risk factor for VTE and ischemic stroke, which are pathological conditions that occur at the early stages of MS [11,43]. Since fibrinogen deposition in the CNS of MS animal models has been shown to affect microglial activity, cause abnormal BBB permeability that contributes to T lymphocyte infiltration, and enhance the interplay between coagulation and inflammation in pre-demyelinated regions [49], the beta-fibrinogen 455 G>A mutation should be further investigated with respect to the content of fibrinogen accumulation, since 455 G>A mutation associated with elevated plasma fibrinogen levels.

Importantly, the underlying mechanisms that are involved in the VTE and CVD onset have also been linked to the presence of the PAI-1 4G/5G polymorphism [16,43]. Although PAI-1 plays an essential role in the fibrinolytic system by inhibiting tPA, hence suppressing plasmin formation, only a few studies have examined the impact of the PAI-1 4G/5G polymorphism on the coagulation–neuroinflammation circuit [50]. Research on CVD, especially on MI, has demonstrated that the 4G allele is a high-risk factor for disease onset [6], whereas in patients with coronary artery disease, the 4G allele was detected more frequently than the 5G allele [51,52]. On the contrary, the 4G allele seems to have a neuroprotective effect against cerebral ischemia, whereas the 5G allele and 5G/5G genotype have been correlated with hemorrhage in post-lysis stroke patients [53]. In the current study, we achieved to show that the 5G allele and 5G/5G genotype were associated with MS susceptibility, demonstrating an increased prevalence compared with healthy participants, which can serve as a predictor risk factor for the occurrence of the disease. Similarly, a previous study on MS found that 5G/5G was associated with harmful effects on MS susceptibility, and the genotype PAI-1 5G/5G by itself or together with the tPA I allele could be crucial risk factors for MS onset [16].

No significant difference in the genotype distribution was observed between MS patients and controls when analyzing MTHFR 677 C>T and 1298 A>C polymorphisms. According to the results of the odds analysis, there was no significant probability of correlating the genotypes with the onset of MS. Previous research on MS showed that the MTHFR 1298 A>C variant was more likely to be associated with disease onset than the MTHFR 6777 C>T variant [54]. Nonetheless, other studies have not found any correlation between mutations and MS, but this may be due to a small study population and different ethnicity, which could account for a different genetic makeup between the study groups [55,56]. In a comprehensive analysis regarding sex, we discovered that men who carried the mutant allele had a greater likelihood of experiencing worse MS progression, but female carriers were not affected. In addition to confirming that disease progression is more severe in male MS patients than in female patients, our results indicate that genetic markers not only may serve as potential biomarkers for the disease but also could be predictive of disease worsening or advanced disability in male patients [45,46].

ACE I/D and Apo B R3500Q polymorphisms have also been associated with thrombosis and CVD since they increase the risk of MI and thrombotic episodes, whereas the ACE D/D variant has also been linked to elevated plasma levels of ACE and suppression of fibrinolysis [57,58]. In the current study, we observed similar frequencies of the ACE I/D and Apo B R3500Q variants between MS patients and healthy controls, and our findings were in agreement with previous studies in the coronary artery that did not demonstrate any correlation between disease occurrence and such mutations [29].

The current research project has some limitations that warrant discussion. Considering the small sample size of participants and the fact that they represent only one population, the statistical power may be diminished, affecting the validity of substantial conclusions. Moreover, the study was limited to patients who tested positive for IgG antibodies against seven coagulant serine proteases, i.e., FVIIa, thrombin, prothrombin, FXa, FXII, plasmin, and protein C. Therefore, further studies are needed to validate our results, recruiting large case-control groups from a wide range of ethnicities to shed more light on the context of gene–gene interactions, combined mutations, and molecular mechanisms that play a crucial role in the coagulation–inflammation interplay. In addition, a comparison of patients who tested positive for antibodies against serine proteases of the coagulation cascade and those who were negative will provide further data on neuroinflammation and neurodegeneration regarding the genetic polymorphisms related to coagulation or thrombosis. Furthermore, the present study focused on identifying high-risk polymorphisms in MS patients. To characterize the role of such polymorphisms and gain a better understanding of the mechanisms that underlie MS pathology, preclinical studies should also be considered. Genetic polymorphisms related to thrombosis could serve as biomarkers for disease prognosis and monitoring and could also be a potential target in therapeutic strategies.

## 5. Conclusions

In conclusion, compelling evidence was revealed when assessing MS occurrence with genetic markers related to thrombophilia and CVD. Among the eleven polymorphisms studied, the homozygous PAI-1 5G/5G genotype and PAI-1 5G allele were found to play a role in MS, whereas the FXIII Val34Leu polymorphism demonstrated a protective role. According to the best of our knowledge, this is the first study to examine such polymorphisms and disease outcomes, finding a positive association between HPA-1a/1b and advanced disability as well as disease severity. The results of an in-depth analysis further revealed the differences regarding sex, demonstrating that disability was associated with male patients carrying the HPA-1b allele, whereas female carriers of HPA-1b were more likely to be correlated with an increase in MS severity. The severity of MS was also significantly increased in male patients carrying the mutant allele of MTHFR 1298 A>C but not in female carriers. Consequently, our findings emphasize the importance of genetics in the MS population and the impact that different genotypes may have on patients’ clinical outcomes.

## Figures and Tables

**Table 1 biomedicines-10-02665-t001:** Demographic, clinical, and laboratory features of MS patients and healthy controls.

Features	MS Patients (*n* = 48)	HCs (*n* = 25)
Sex		
Women/Men	28/20	16/9
Age in years		
Mean ± S.D.	49.5 ± 13.8	40.7 ± 10.0
Min–max	23–80	21–55
Disease course (RRMS/SPMS/PPMS)	39/8/1	N/A
Disease duration		
Mean ± S.D.	15.52 ± 8.5	N/A
Median (interquartile range)	16 (10–21)	
EDSS		
Median (interquartile range, IQR)	3.25 (2.5–5.5)	N/A
Mild: 0–3.0 [n (%)]	24 (50.0)	
Moderate: 3.5–5.5 [n (%)]	17 (35.4)	
Severe: 6.0–9.5 [n (%)]	7 (14.6)	
Women/Men (median, IQR)	3.0 (2.5–4.75)/4.0 (2.5–6.0)	
MSSS		
Median (interquartile range)	3.65 (2.64–5.41)	N/A
Benign MS: 0.01–1.99 [n (%)]	7 (14.6)	
Moderate MS: 2.00–6.99 [n (%)]	36 (75.0)	
Severe MS: 7–10 [n (%)]	4 (8.3)	
N/A	1 (2.1)	
Women/Men (median, IQR)	3.14 (2.35–5.48)/4.20 (2.82–5.41)	
Laboratory findings		
Anti-FVIIa [n (%)]	11 (22.9)	N/A
Anti-thrombin [n (%)]	6 (12.5)	
Anti-prothrombin [n (%)]	7 (14.6)	
Anti-FXa [n (%)]	11 (22.9)	
Anti-FXII [n (%)]	9 (18.7)	
Anti-plasmin [n (%)]	15 (31.2)	
Anti-protein C [n (%)]	6 (12.5)	

MS: multiple sclerosis; HC: healthy controls; S.D.: standard deviation; RRMS: relapsing–remitting multiple sclerosis; SPMS: secondary progressive multiple sclerosis; PPMS: primary progressive multiple sclerosis; EDSS: expanded disability status scale; MSSS: multiple sclerosis severity score; N/A: not applicable.

**Table 2 biomedicines-10-02665-t002:** Selected risk factors determined by sequence-specific oligonucleotides.

Protein	Gene	Mutation and Polymorphism	Ref SNP
Blood coagulation factor V	F5	1691 G>A (Leiden)	rs6025
		His1299Arg (HR2 haplotype)	rs1800595
Prothrombin (Blood coagulation factor II)	F2	20210 G>A	rs1799963
Blood coagulation factor XIII	F13A1	Val34Leu	rs5985
ß-Fibrinogen	FGB	−445 G>A	rs1800790
Human platelet antigen 1 (HPA1)	ITGB3	1a/1b (Leu33Pro)	rs5918
5,10-Methylenetetrahydrofolate reductase	MTHFR	677 C>T	rs1801133
		1298 A>C	rs1801131
Plasminogen activator inhibitor 1 (PAI-1)	Serpine1	4G/5G	rs1799762
Angiotensin-converting enzyme (ACE)	ACE	I/D (Insertion or Deletion)	rs1799752
Apolipoprotein B (Apo B)	ApoB	Arg3500Gln	rs5742904

SNP: single nucleotide polymorphism; His: histidine; Arg: arginine; Val: valine; Leu: leucine; Pro: proline; Gln: glutamine; Cys: cysteine.

**Table 3 biomedicines-10-02665-t003:** Genotype frequencies distributions of thrombophilia factors performed by sequence-specific oligonucleotide probing in MS patients and healthy controls.

Risk Factors	Genotype	MS (n (%))	HC (n (%))	** p* Value	Hardy–Weinberg Equilibrium (Only for HCs)Chi-Square (χ^2^)	Odds Ratio (95% CI)
Factor V Leiden	G/G	39 (81.25)	22 (88.0)	0.53	0.10	Ref
	G/A	9 (18.75)	3 (12.0)			1.69 (0.41–6.91)
	A/A	0	0	-		-
	G/A + A/A	9 (18.75)	3 (12.0)	0.53		1.69 (0.41–6.91)
Factor V R2	A/A	40 (83.33)	21 (84.0)	1.00	0.19	Ref
	A/G	8 (16.67)	4 (16.0)			1.05 (0.28–3.90)
	G/G	0	0	-		-
	A/G + G/G	8 (16.67)	4 (16.0)	1.00		1.05 (0.28–3.90)
Factor II	G/G	47 (97.92)	25 (100)	1.00	-	Ref
	G/A	1 (2.08)	0			1.61 (0.06–41.01)
	A/A	0	0	-		-
	G/A + A/A	1 (2.08)	0	1.00		1.61 (0.06–41.01)
Factor FXIII Val34Leu	G/G	34 (70.83)	12 (48.0)	-	0.91	Ref
	G/T	12 (25.0)	12 (48.0)	0.04 *		0.35 (0.12–0.99)
	T/T	2 (4.17)	1 (4.0)	1.00		0.70 (0.05–8.51)
	G/T + T/T	14 (29.17)	13 (52.0)	0.055		0.38 (0.14–1.03)
β-Fibrinogen	G/G	23 (47.92)	16 (64.0)	-	1.20	Ref
	G/A	21 (43.75)	9 (36.0)	0.34		1.62 (0.59–4.45)
	A/A	4 (8.33)	0	0.27		6.32 (0.32–125.6)
	G/A + A/A	25 (52.08)	9 (36.0)	0.19		1.93 (0.71–5.22)
HPA-1	1a/1a	34 (70.83)	17 (68.0)	-	0.06	Ref
	1a/1b	14 (29.17)	7 (28.0)	1.00		1.00 (0.34–2.94)
	1b/1b	0	1 (4.0)	0.34		0.17 (0.0–4.37)

* Chi-square test and, when necessary, Fisher’s exact tests were used to determine genotype and allele frequency distributions, whereas logistic regression analysis was applied to evaluate possible associations. MS: multiple sclerosis; HC: healthy controls; HPA-1: human platelet antigen 1. Statistically significant values *p* < 0.05 (*).

**Table 4 biomedicines-10-02665-t004:** Genotype frequencies distributions of cardiovascular risk factors performed by sequence-specific oligonucleotide probing in MS patients and healthy controls.

Risk Factors	Genotype	MS (n (%))	HC (n (%))	** p* Value	Hardy–Weinberg Equilibrium (Only for HCs) Chi-Square (χ^2^)	Odds Ratio (95% CI)
PAI-1	4G/4G	8 (16.67)	8 (32.0)	-	0.69	Ref
	4G/5G	21 (43.75)	14 (56.0)	0.50		1.50 (0.46–4.93)
	5G/5G	19 (39.58)	3 (12.0)	0.02 *		6.33 (1.32–30.24)
	4G/4G + 4G/5G	29 (60.42)	22 (88.0)	-		Ref
	5G/5G	19 (39.58)	3 (12.0)	0.016 *		4.80 (1.26–18.31)
MTHFR (C677T)	C/C	18 (37.5)	12 (48.0)	-	0.98	Ref
	C/T	25 (52.08)	9 (36.0)	0.25		1.85 (0.64–5.32)
	T/T	5 (10.42)	4 (16.0)	1.00		0.83 (0.18–3.75)
	C/T + T/T	30 (62.50)	13 (52.0)	0.38		1.54 (0.57–4.09)
MTHFR (A1298C)	A/A	21 (43.75)	10 (40.0)	-	0.04	Ref
	A/C	23 (47.92)	12 (48.0)	0.86		0.91 (0.32–2.55)
	C/C	4 (8.33)	3 (12.0)	0.67		0.63 (0.12–3.39)
	A/C + C/C	27 (56.25)	15 (60.0)	0.76		0.85 (0.32–2.29)
ACE	I/I	3 (6.25)	4 (16.0)	-	0.98	Ref
	I/D	26 (54.17)	9 (36.0)	0.17		3.85 (0.72–20.63)
	D/D	19 (39.58)	12 (48.0)	0.42		2.11 (0.40–11.13)
Apo B	G/G	48(100)	25 (100)	-	-	-
	G/A	0	0			
	A/A	0	0			

* Chi-square test and, when necessary, Fisher’s exact tests were used to determine genotype and allele frequency distributions, whereas logistic regression analysis was applied to evaluate possible associations. MS: multiple sclerosis; HC: healthy controls; PAI-1: plasminogen activator inhibitor-1; MTHFR: methylenetetrahydrofolate reductase; ACE: angiotensin-converting enzyme; Apo: apolipoprotein. Statistically significant value *p* < 0.05 (*).

**Table 5 biomedicines-10-02665-t005:** Allele frequencies distributions of thrombophilia factors performed by sequence-specific oligonucleotide probing in MS patients and healthy controls.

Risk Factors	Allele	MS (%)	HC (%)	** p* Value	Odds Ratio (95% CI)
Factor V Leiden	G	87 (90.62)	47 (94.0)	0.75	Ref 1.62 (0.42–6.28)
	A	9 (9.38)	3 (6.0)		
Factor V R2	A	88 (91.67)	46 (92.0)	1.00	Ref 1.04 (0.30–3.66)
	G	8 (8.33)	4 (8.0)		
Factor II	G	95 (98.96)	50 (100)	1.00	Ref 1.59 (0.06–39.69)
	A	1 (1.04)	0		
Factor FXIII Val34Leu	G	80 (83.33)	36 (72.0)	0.10	Ref 0.51 (0.22–1.17)
	T	16 (16.67)	14 (28.0)		
beta- Fibrinogen	G	67 (69.80)	41 (82.0)	0.11	Ref 1.97 (0.85–4.58)
	A	29 (30.20)	9 (18.0)		
HPA-1	1a	82 (85.42)	41 (82.0)	0.59	Ref 0.77 (0.31–1.95)
	1b	14 (14.58)	9 (18.0)		

* Chi-square test and, when necessary, Fisher’s exact tests were used to determine genotype and allele frequency distributions, whereas logistic regression analysis was applied to evaluate possible associations. MS: multiple sclerosis; HC: healthy controls; HPA-1: human platelet antigen 1.

**Table 6 biomedicines-10-02665-t006:** Allele frequencies distributions of cardiovascular risk factors performed by sequence-specific oligonucleotide probing in MS patients and healthy controls.

Risk Factors	Allele	MS (%)	HC (%)	** p* Value	Odds Ratio (95% CI)
PAI-1	4G	37 (38.54)	30 (60.0)	0.013 *	Ref 2.39 (1.19–4.81)
	5G	59 (61.46)	20 (40.0)		
MTHFR (C677T)	C	61 (63.54)	33 (66.0)	0.77	Ref 1.11 (0.54–2.28)
	T	35 (36.46)	17 (34.0)		
MTHFR (A1298C)	A	65 (67.71)	32 (64.0)	0.65	Ref 0.85 (0.41–1.74)
	C	31 (32.29)	18 (36.0)		
ACE	I	32 (33.33)	17 (34.0)	0.93	Ref 1.03 (0.50–2.12)
	D	64 (66.67)	33 (66.0)		
Apo B	G	96 (100)	50 (100)	-	-
	A	0	0		

* Chi-square test and, when necessary, Fisher’s exact tests were used to determine genotype and allele frequency distributions, whereas logistic regression analysis was applied to evaluate possible associations. MS: multiple sclerosis; HC: healthy controls; PAI-1: plasminogen activator inhibitor-1; MTHFR: methylenetetrahydrofolate reductase; ACE: angiotensin-converting enzyme; Apo: apolipoprotein. Statistically significant value *p* < 0.05 (*).

**Table 7 biomedicines-10-02665-t007:** Association of thrombophilia factor genotype with EDSS and MSSS scores.

Risk Factors	Genotype	EDSS			MSSS		
		Median (IQR)	Odds Ratio (95% CI)	*p* Value for OR	Median (IQR)	Odds Ratio (95% CI)	*p* Value for OR
Factor V Leiden	G/G	3.25 (2.50–5.50)	Ref		3.65 (2.64–5.41)	Ref	
	G/A	3.00 (1.75–3.50)	0.53 (0.25–0.90)	0.016 (*)	2.73 (1.90–3.93)	0.70 (0.44–1.06)	0.10
Factor V R2	A/A	3.25 (2.50–5.50)	Ref		3.65 (2.64–5.41)	Ref	
	A/G	1.75 (1.00–4.75)	0.46 (0.19–0.84)	0.008 (**)	2.10 (1.45–4.49)	0.60 (0.30–0.97)	0.038 (*)
Factor FXIII Val34Leu	G/G	3.50 (2.50–5.50)	Ref		3.65 (2.64–5.60)	Ref	
	G/T, T/T	3.00 (2.50–5.75)	1.02 (0.71–1.44)	0.90	4.16 (2.55–6.28)	1.16 (0.87–1.57)	0.31
beta-Fibrinogen −455 G>A	G/G	3.50 (2.50–5.50)	Ref		3.65 (2.50–5.41)	Ref	
	G/A, A/A	3.50 (2.50–5.50)	0.99 (0.72–1.37)	0.99	4.21 (3.02–5.26)	1.06 (0.81–1.40)	0.66
HPA-1	1a/1a	3.25 (2.50–5.50)	Ref		3.65 (2.64–5.41)	Ref	
	1a/1b	4.50 (2.87–6.12)	1.47 (1.03–2.18)	0.03 (*)	4.54 (3.08–6.47)	1.42 (1.05–2.01)	0.02 (*)

EDSS: expanded disability status scale; MSSS: multiple sclerosis severity score; IQR: interquartile range; HPA-1: human platelet antigen 1. Statistically significant values *p* < 0.05 (*), *p* < 0.01 (**).

**Table 8 biomedicines-10-02665-t008:** Association of cardiovascular risk factor genotype with EDSS and MSSS scores.

Risk Factors	Genotype	EDSS			MSSS		
		Median (IQR)	Odds Ratio (95% CI)	*p* Value for OR	Median (IQR)	Odds Ratio (95% CI)	*p* Value for OR
PAI–1	4G/4G	4.00 (2.75–5.50)	Ref		3.65 (2.40–5.63)	Ref	
	4G/5G, 5G/5G	3.00 (2.50–5.12)	0.79 (0.52–1.20)	0.27	3.50 (2.64–5.03)	0.88 (0.62–1.25)	0.46
MTHFR (C677T)	CC	3.50 (2.50–5.50)	Ref		3.69 (2.64–5.50)	Ref	
	CT, TT	3.50 (2.50–5.50)	1.05 (0.75–1.47)	0.78	3.57 (2.30–5.12)	0.92 (0.70–1.23)	0.60
MTHFR (A1298C)	AA	3.50 (2.50–5.50)	Ref		3.65 (2.64–5.03)	Ref	
	AC, CC	3.00 (2.50–5.50)	1.09 (0.79–1.52)	0.60	4.25 (2.77–6.45)	1.33 (0.99–1.87)	0.055
ACE	I/I	3.00 (2.50–5.50)	Ref		3.50 (2.57–4.64)	Ref	
	I/D, D/D	3.50 (2.50–5.50)	0.81 (0.43–1.58)	0.52	3.57 (2.53–4.98)	0.69 (0.39–1.16)	0.16

EDSS: expanded disability status scale; MSSS: multiple sclerosis severity score; IQR: interquartile range; PAI-1: plasminogen activator inhibitor-1; MTHFR: methylenetetrahydrofolate reductase; ACE: angiotensin-converting enzyme; Apo: apolipoprotein.

**Table 9 biomedicines-10-02665-t009:** Sex-based association of genetic polymorphisms that were correlated with increased EDSS and MSSS scores.

Risk Factors	Genotype	EDSS			MSSS		
		Median (IQR)	Odds Ratio (95% CI)	*p* Value for OR	Median (IQR)	Odds Ratio (95% CI)	*p* Value for OR
HPA-1 (Women)	1a/1a	3.00 (2.50–4.00)	Ref		2.64 (1.60–4.25)	Ref	
	1a/1b	4.00 (2.50–5.50)	1.54 (0.90–2.99)	0.11	4.49 (3.01–6.24)	1.56 (1.04–2.61)	0.03 (*)
HPA-1 (Men)	1a/1a	3.50 (2.25–5.50)	Ref		3.69 (2.68–4.98)	Ref	
	1a/1b	6.50 (6.00–8.00)	3.04 (1.22–19.54)	0.01 (*)	6.43 (4.5408.47)	1.63 (0.93–3.35)	0.08
MTHFR (A1298C) (Women)	AA	3.50 (3.00–4.00)	Ref		2.97 (2.30–3.79)	Ref	
	AC, CC	3.00 (2.50–5.25)	0.97 (0.57–1.70)	0.93	4.21 (2.47–6.01)	1.23 (0.84–1.92)	0.29
MTHFR (A1298C) (Men)	AA	3.25 (2.37–5.62)	Ref		3.62 (2.42–4.87)	Ref	
	AC, CC	4.00 (2.62–7.12)	1.21 (0.80–1.91)	0.37	4.38 (3.35–8.35)	1.66 (1.02–3.43)	0.04 (*)

EDSS: expanded disability status scale; MSSS: multiple sclerosis severity score; IQR: interquartile range; HPA-1: human platelet antigen 1; MTHFR: methylenetetrahydrofolate reductase. Statistically significant values *p* < 0.05 (*).

## Data Availability

The data that support the findings of this study are available from the corresponding author upon reasonable request.

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
