# Peer review of "Genetic Markers for Thrombophilia and Cardiovascular Disease Associated with Multiple Sclerosis"

_biomedicines, 2022, doi:10.3390/biomedicines10102665_

Round 1

Reviewer 1 Report

The article entitled Genetic markers for thrombophilia and cardiovascular disease associated with Multiple Sclerosis is a document of interesting subject matter.

1. It is expected to have an extensive literature review followed by an in-depth and critical analysis of the state of the art, and identify challenges for future research in the Introduction.

2. The authors should do the analysis the conclusion section must clearly establish a strong correlation with the proposed topic.
3. Your abstract should clearly state the essence of the problem you are addressing, what you did and what you found and recommend. That will help a prospective reader of the abstract to decide if they wish to read the entire article

4. The objective or objectives should be clearly elucidated in the last paragraph of the introduction.

Author Response

The article entitled Genetic markers for thrombophilia and cardiovascular disease associated with Multiple Sclerosis is a document of interesting subject matter.

  1. It is expected to have an extensive literature review followed by an in-depth and critical analysis of the state of the art, and identify challenges for future research in the Introduction.

We have revised our Introduction section with a more extensive review of the literature, as given on pages 4-7, beginning on page 4 with the sentence that reads: “One of the factors associated with VTE is a point mutation in the gene encoding the factor V (FV) that substitutes guanine for adenine (G>A) at nucleotide 1691, resulting in a missense mutation, also known as FV Leiden R506Q” until page 7 with the sentence that reads: “Moreover, we investigated whether such polymorphisms can serve as predictors of advanced disability and progression of the disease.”

  1. The authors should do the analysis the conclusion section must clearly establish a strong correlation with the proposed topic.

We have amended the discussion section, assessing in more detail all the genetic polymorphisms studied in the current work. Specifically, we emphasized each polymorphism separately, giving our results an in-depth analysis and comparing them with data published in previous studies, providing thought for future research on thrombosis-neuroinflammation cross-talk and genetic predisposition. The revisions in the discussion section begin on page 11 with the line that reads: “In this case-control study...” until page 17: “Genetic polymorphisms related to thrombosis could serve as biomarkers for disease prognosis and monitoring and could also be a potential target in therapeutic strategies.”

Furthermore, we included a conclusion section describing all the significant findings of our work in summary (page:18).

  1. Your abstract should clearly state the essence of the problem you are addressing, what you did and what you found and recommend. That will help a prospective reader of the abstract to decide if they wish to read the entire article.

Folliwing the reviewer’s suggestion, the abstract has been modified as follows: “Multiple Sclerosis (MS) is an autoimmune inflammatory disease of the central nervous system (CNS) with an unknown etiology, although genetic, epigenetic, and environmental factors are thought to play a role. Recently, coagulation components have been shown to provide immunomodulatory and pro-inflammatory effects in the CNS, leading to neuroinflammation and neurodegeneration. The current study aimed to determine whether patients with MS exhibited an overrepresentation of polymorphisms implicated in the coagulation and whether such polymorphisms are associated with advanced disability and disease progression. The cardiovascular disease (CVD) strip assay was applied to 48 MS patients and 25 controls to analyze 11 genetic polymorphisms associated with thrombosis and CVD. According to our results, FXIIIVal34Leu heterozygosity was less frequent [OR:0.35 (95% CI: 0.12-0.99);p=0.04], whereas PAI-1 5G/5G homozygosity was more frequent in MS [OR:6.33 (95% CI: 1.32-30.24);p=0.016]. In addition, carriers of the HPA-1a/1b were likely to have advanced disability [OR:1.47 (95% CI: 1.03-2.18);p=0.03] and disease worsening [OR:1.42 (95% CI: 1.05-2.01);p=0.02]. A gender-based analysis revealed that male HPA-1a/1b carriers were associated with advanced disability [OR:3.04 (95% CI: 1.22-19.54);p=0.01], while female carriers had an increased likelihood of disease worsening [OR:1.56 (95% CI: 1.04-2.61);p=0.03]. Our findings suggest that MS may be linked to thrombophilia-related polymorphisms, which warrants further investigation.”

  1. The objective or objectives should be clearly elucidated in in the last paragraph of the Introduction.

To better clarify our objectives of the present study, we changed the last paragraph of the Introduction on page 6-7 as follows: “Given the significance of variants in genes encoding molecules involved in the coagulation cascade and the importance of coagulation-inflammation interplay in neuroinflammatory and neurodegenerative diseases, the purpose of our research was intended to examine such variants in MS. Specifically, we aimed to examine whether patients with MS displayed overrepresentations in the following genetic markers: FV Leiden polymorphism, FV R2, Prothrombin 20210G>A, FXIII Val34Leu, beta-fibrinogen -455G>A, HPA1a/1b, MTHFR 677 C>T, MTHFR 1298 A>T, PAI-1 4G/5G, and ACE I/D compared to healthy individuals. Moreover, we investigated whether such polymorphisms can serve as predictors of advanced disability and progression of the disease.”

Reviewer 2 Report

The short communication of Hadjiagapiou et al. aims to identify thrombophilia and cardiovascular disease-associated genetic variants linked to multiple sclerosis (MS). To achieve this goal, Authors conducted a clinical trial comparing the frequency of selected genetic variations between MS patients and control subjects. Authors identified one gene SNP (FXIII) with less, and one (PAI-1) with higher occurrence in MS patients. In addition, based upon the available detailed clinical data, Authors found the HPA-1b gene that was associated with unfavourable disability status and MS score compared to the HPA-1a allele. Based upon their investigations, Authors conclude that the identified genes further support previous studies establishing a relationship between coagulation abnormalities and neurogenerative diseases, and further studies are warranted to unravel the underlying molecular mechanisms.  

Authors complement the manuscript with 8 detailed Tables and cite 26 relevant publications to describe current knowledge in the field and to put their results into context.

The topic fits the scope of “Biomolecules” and is of interest for its readers. The text is succinct and written in a clear fashion, the Methods provide all necessary information, in particular the issues related to ethical concerns.

This reviewer notes these issues that should be addressed by the Authors:

1.     Table 1: in the heading the number of MS patients are marked as 50, however in the statistics are only 48 are used. Is it a mistyping or there were some patients that were later excluded from the study? 

It is an important issue as some % values in the Table refer to n=50 (MSSS and Laboratory findings) while the EDSS % values refer to 48 patients. 

Please revise all Tables for accuracy, too.

2.     Authors should also revise their data for accuracy concerning gene allele frequency results.

3.     Interestingly, the Female/Male ratio in the recruited MS patients is 28/20 (approx.. 1.4 times more females than males). In general, MS frequency in females is approx.2.5-3 times higher than in males. Can Authors comment in this issue? What is known about the frequency of IgG antibodies, that was used as selection criteria, between females and males? What are the current statistics concerning sex differences of MS patients in Cyprus in particular?

4.      A recent study suggested that disease progression, in particular grey matter thinning is more severe in male MS patients compared to females (see: Voskuhl et al. Biology of Sex Differences (2020) 11:49).  Did Authors attempt to analyse the HPA-1b gene allele frequency that they found to be related to MS severity score in male patients exclusively? 

5.     The number of patients included in the study is modest and represents only one population. Authors should mention this limitation in the discussion part.

6.     Authors should provide detailed, individualized data of patients in a Supplementary Table.

7.     Table 8: There is a technical problem with the Table with overlapping/interfering lines, please correct.

Author Response

  1. Table 1: in the heading the number of MS patients are marked as 50, however in the statistics are only 48 are used. Is it a mistyping or there were some patients that were later excluded from the study?

It is important issue as some % values in the Table refer to n= 50 (MSSS and Laboratory findings) while the EDSS % values refer to 48 patients. Please revise all Tables for accuracy, too.

As the reviewer correctly pointed out, the number of MS patients who participated in the present study was 48 individuals. The present study included patients who had antibodies against the coagulation factors studied in our previously published work (Hadjiagapiou et al., Antibodies to blood coagulation components are implicated in patients with multiple sclerosis (2022)) since we were interested in further investigating the role of the blood coagulation constituents. The polymorphisms in the two patients included in our previous work, who met the inclusion criteria for the present study were previously analyzed. However, since these were not analyzed for all the polymorphisms studied, we have now excluded them. Therefore, all the Tables have been corrected with the correct number of MS participants (n = 48).

The range of values of each category (benign, moderate, and severe), as well the number of patients correlated to the different MSSS scores: (Table 1) Benign MS: 0.01-1.99 n= 7 (14.6%), Moderate MS: 2.00-6.99 n= 36 (75.0%), Severe MS: 7-10 n= 4 (8.3%), and N/A n=1 (2.1%) are presented more clearly. The disease duration of the last patient was 0 years, so according to the MSSS provided by Roxburgh et al. (Roxburgh et al., Multiple Sclerosis Severity Score: using disability and disease duration to rate disease severity (2005), 64(7)), there is no particular value corresponding to 0 years of disease duration.

In section 3.1 of the Results (Demographic and clinical data of study participants), we corrected the percentage of MS patients who experienced aggressive disease progression (8.3%) (pages 8-9).

In Table 1 we have also corrected the number of patients (n = 48) and the percentages of the laboratory findings according to our previous work (Hadjiagapiou et al, 2022): anti-FVIIa n=11 (22.9%), anti-thrombin n=6 (12.5%), anti-prothrombin n=7 (14.6%), anti-FXa n=11 (22.9%), anti-FXII n=9 (18.7%), anti-plasmin n=15 (31.2%), and anti-protein C n=6 (12.5%). As can be seen from the Supplementary data, there were MS patients that were positive for more than one IgG antibody.

  1. Authors should also revise their data for accuracy concerning gene allele frequency results.

All the tables have been revised for accuracy, using the correct number of MS patients in the current study (n=48). Therefore, we corrected the number of patients harbouring the Apo B GG genotype in Table 4 to 48 (page 28), and the frequency of the Apo B G allele in Table 6 to 96 (page 30).

  1. Interestingly, the Female/Male ratio in the recruited MS patients in 28/20 (approx..1.4 times more females than males). Can authors comment in this issue? What is known about the frequency of IgG antibodies that was used as selection criteria between females and males? What are the current statistics concerning sex differences of MS patients in Cyprus in particular?

The present study follows our previously published work ((Hadjiagapiou et al., Antibodies to blood coagulation components are implicated in patients with multiple sclerosis (2022)), which aimed to analyze the cross-talk between coagulation and inflammation in MS through the detection of antibodies against coagulation components and to investigate how such molecules affect the disability status and progression of the disease. In that study, we analyzed samples from 167 MS patients and found that 72 (43%) were positive for the presence of the antibodies studied (anti-FVIIa, anti-thrombin, anti-prothrombin, anti-FXa, anti-FXII, anti-plasmin, and anti-protein C).   

The purpose of the present study was to detect/determine the frequency of genetic polymorphisms associated with thrombophilia in MS and investigate their role in disease progression. The study was conducted using samples from the 72 MS patients who tested positive for the antibodies against clotting factors in our previous work (Hadjiagapiou et al., 2022) in order to gain a better understanding of the coagulation/thrombosis in neuroinflammatory diseases. We sampled 28 females and 20 males in this study due to limited funding and consent of a certain number of patients. The ratio of women/men, as correctly pointed out by the reviewer was 1.4:1.

The most recent study in Cyprus that documented the female-to-male ratio in MS was carried out by Charalambidou et al. (Charalambidou et al., Multiple Sclerosis in Cyprus: A Fourteen Year (2000-2014), Epidemiological Study (2016) 4:1), revealing a female/male ratio of 1.6:1 in the MS Cypriot population.

Regarding the presence of IgG antibodies against coagulation components between men and women in MS, our study (Hadjiagapiou et al. Antibodies to blood coagulation components are implicated in patients with multiple sclerosis (2022)) was the first to evaluate these antibodies in MS. According to our odds ratio analysis there was no significant correlation between antibodies and gender: (males: ref) [anti-FVIIa OR: 0.68 (95% CI: 0.26-1.86); anti-thrombin OR: 2.22 (95% CI: 0.54-15.0); anti-prothrombin OR: 0.93 (95% CI: 0.27-3.67); anti-FXa OR: 1.78 (95% CI: 0.59-6.55); anti-FXII OR: 0.59 (95% CI: 0.17-2.19); anti-plasmin OR: 0.59 (95% CI: 0.23-1.56); anti-protein C OR: 3.54 (95% CI: 0.58-68.2)].

Moreover, previous studies analyzing antibodies against clotting factors in diseases with clinical features similar to MS, like for example the antiphospholipid syndrome, did not mention any difference regarding gender (e.g. B. Artim-Esen et al., Anti-factor Xa antibodies in patients with antiphospholipid syndrome and their effects upon coagulation assays, (2015); 17(1):47).

  1. A recent study suggested that disease progression, in particular grey matter thinning is more severe in male MS patients compared to females (see: Voskuhi et al. Biology of sex differences (2020) 11:49). Did Authors attempt to analyse the HPA-1b gene allele frequency that they found to be related to MS severity score in male patients exclusively?

Considering the proposed study of Voskuhi et al., as well as a another recent study (Coyle: What Can We Learn from Sex Differences in MS? (2021) 11:10), in which researchers claim that there was significant worsening annually on EDSS compared to females, we firstly analyzed the median indices and interquartile range of male and female MS patients separately, observing an increase median index in male patients compared to females in both EDSS and MSSS scores (Table 1, page 25). This observation is now mentioned in section 3.1 (Demographic and clinical data of study participants, page 9), which reads: “A gender-specific analysis of disability status and MS severity score revealed an increase in EDSS and MSSS in males compared to females. In particular, male patients showed an EDSS median of 4.00 (Interquartile range, IQR: 2.5-6.0) and MSSS median of 4.20 (IQR: 2.82-5.41), whereas female patients had an EDSS median of 3.0 (IQR: 2.5-4.75) and MSSS of 3.14 (IQR: 2.35-5.48).”

Subsequently, the reviewer asked us to analyse the possible association of the HPA-1a/1b genotype frequency in regard to gender (males and females), hence Table 9 is now included on page 33 which evaluates the potential association of the HPA-1a/1b regarding the EDSS and MSSS scores in males and females, separately. We mentioned this observation in section 3.3 (Mutant allele presence and disability progression in MS, page 10) as follows: “A detailed analysis of HPA-1a/1b genotype frequencies indicated that there was a significant difference between male and female patients with MS. As shown in Table 9, male HPA-1a/1b carriers were significantly more likely to have a high EDSS score [OR:3.04 (1.22-19.54); p<0.05), whereas female carriers showed no significant association. On the other hand, female MS patients carrying the HPA-1a/1b genotype had an increased likelihood to be correlated with an increased MSSS score [OR: 1.56 (1.04-2.61), p<0.05]; however, such a correlation was not observed in the group of male patients.” 

We also proceeded with a detailed analysis for all polymorphisms studied based on gender to identify any association between the presence of a particular genetic polymorphism and the EDSS and/or MSSS, and surprisingly, we found that MTHFR A1298C showed significant results when analysing males and females separately (Table 9, page 33). In section 3.3 (page 10), we now report this observation: “Another important observation was revealed when analysing the MTHFR A1298C polymorphism. Although there was no association between heterozygous or homozygous genotypes for the MTHFR A1298C polymorphism and the clinical outcomes of the disease, further analysis in male patients carrying the mutant C allele demonstrated an increased likelihood of having worse MS progression [OR: 1.66 (1.02-3.43), p<0.05]; a finding that was not observed in female patients (Table 9).

Additionally, we discussed our findings in the discussion section, citing relevant literature as well: (pages13-14) “Recent studies highlighted that male MS patients experienced ... In view of these findings, it is increasingly clear that gender plays a role in MS severity, even in genetic predispositions that show significant gender differences.” And (pages 15-16) “In a comprehensive analysis regarding gender, we discovered that men who carried the mutant allele ... our results indicate that genetic markers may not only serve as potential biomarkers for the disease but could also be predictive of disease worsening or advanced disability in males.”    

At this point we will like to mention the revision of Tables 7 and 8 according to our statistician's suggestion (pages 31-32). Since the analysis for the possible association between EDSS/MSSS and the presence of the mutant allele was done considering both the heterozygous and the homozygous genotype with the mutant allele, the recommendation is to clearly present the genotypes assessed in each case.

  1. The number of patients included in the study is modest and represents only one population. Authors should mention this limitation in the discussion part.

The reviewer suggests to acknowledge the fact that the patients studied represent only one population. This is now included in the last paragraph on page 17, acknowledging the limitations of our study and now reads: “The current research project has some limitations that warrant discussion… Genetic polymorphisms related to thrombosis could serve as biomarkers for disease prognosis and monitoring and could also be a potential target in therapeutic strategies.”

  1. Authors should provide detailed, individualized data of patients in a Supplementary Table.

We have created Supplementary Tables (1-5), in which we included all the detailed data of our participants (age, gender, disease course, disease duration, type of MS, IgG activity against coagulation components, all the genetic polymorphisms studied in the current study).

  1. Table 8: There is a technical problem with the Table with overlapping/interfering lines, please correct.

All black lines that separated two alleles of a gene have been removed from Table 5, Table 6, Table 7, and Table 8. May we kindly clarify that this is the technical problem that the reviewer is referring to?   

Round 2

Reviewer 2 Report

Authors provided satisfactory answers for the  points raised by this reviewer and revised the manuscript accordingly.

No further issues.